


# Global trends and European emissions of tetrafluoromethane (CF$_4$), hexafluoroethane (C$_2$F$_6$) and octafluoropropane (C$_3$F$_8$)

Daniel Say[1], Alistair J. Manning[2], Luke M. Western[1], Dickon Young[1], Adam Wisher[1], Matthew Rigby[1], Stefan Reimann[3], Martin K. Vollmer[3], Michela Maione[4], Jgor Arduini[4], Paul B. Krummel[5], Jens Mühle[6], Christina M. Harth[6], Brendan Evans[1], Ray F. Weiss[6], Ronald G. Prinn[7], and Simon O'Doherty[1]

[1]Atmospheric Chemistry Research Group, University of Bristol, Bristol BS8 1TS, UK
[2]Met Office Hadley Centre, Exeter EX1 3PB, UK
[3]Empa, Swiss Federal Laboratories for Materials Science and Technology, Ueberlandstrasse 129, 8600, Dübendorf, Switzerland
[4]Department of Pure and Applied Sciences, University of Urbino, Urbino, Italy
[5]Climate Science Centre, CSIRO Oceans and Atmosphere, Aspendale, Australia
[6]Scripps Institution of Oceanography, University of California, San Diego, La Jolla, USA
[7]Center for Global Change Science, Massachusetts Institute of Technology, 77 Massachusetts Ave, Building 54-1312, Cambridge, MA 02139, USA

**Correspondence:** Daniel Say (Dan.Say@bristol.ac.uk)

**Abstract.** Perfluorocarbons (PFCs) are amongst the most potent greenhouse gases listed under the United Nations Framework Convention on Climate Change (UNFCCC). With atmospheric lifetimes in the order of thousands to tens of thousands of years, PFC emissions represent a permanent alteration to the global atmosphere on human timescales. While the industries responsible for the vast majority of these emissions – aluminium smelting and semi-conductor manufacturing – have made efficiency

improvements and introduced abatement measures, the global mean mole fractions of three PFCs, namely tetrafluoromethane (CF$_4$, PFC-14), hexafluoroethane (C$_2$F$_6$, PFC-116) and octafluoropropane (C$_3$F$_8$, PFC-218), continue to grow. In this study, we update baseline growth rates using in-situ high-frequency measurements from the Advanced Global Atmospheric Gases Experiment (AGAGE) and, using data from four European stations, estimate PFC emissions for northwest Europe. The global growth rate of CF$_4$ decreased from 1.3 ppt yr$^{-1}$ in 1979 to 0.6 ppt yr$^{-1}$ around 2010 followed by a renewed steady increase to

0.9 ppt yr$^{-1}$ in 2018. For C$_2$F$_6$, the growth rate grew to a maximum of 0.125 ppt yr$^{-1}$ around 1999, followed by a decline to a minimum of 0.075 ppt yr$^{-1}$ in 2009, followed by weak growth thereafter. The C$_3$F$_8$ growth rate was around 0.007 ppt yr$^{-1}$ until the early 1990s and then quickly grew to a maximum of 0.03 ppt yr$^{-1}$ in 2003/04. Following a period of decline until 2012 to 0.015 ppt yr$^{-1}$, the growth rate slowly increased again to ∼0.017 ppt yr$^{-1}$ in 2019. Unlike CF$_4$ (and to a lesser extent C$_2$F$_6$), we observed no clear minimum associated with the 2008 financial crisis for C$_3$F$_8$. We used an inverse modelling framework to

infer PFC emissions for northwest Europe. No statistically significant trend in regional emissions was observed for any of the PFCs assessed. For CF$_4$, European emissions in early years were linked predominantly to the aluminium industry. However, we link large emissions in recent years to a chemical manufacturer in northwest Italy. Emissions of C$_2$F$_6$ are linked to a range of sources, including a semi-conductor manufacturer in Ireland and a cluster of smelters in Germany's Ruhr valley. In contrast,





northwest European emissions of $C_3F_8$ are dominated by a single source in northwest England, raising the possibility of using

emissions from this site for a tracer release experiment.

## 1   Introduction

Perfluorocarbons (PFCs) – fully fluorinated hydrocarbons – are a group of extremely potent greenhouse gases that are used extensively in the electronics and semi-conductor industry and are also emitted as a by-product of aluminium and rare earth

metal smelting. As a consequence of their high global warming potentials (GWP), PFCs were listed under the Kyoto 'basket', a collection of gases for which regulation on their emissions was introduced under the Kyoto Protocol (UNFCCC, 2009). Countries listed under Annex 1 of the Protocol are required to report annual PFC emissions to the United Nations Framework Convention on Climate Change (UNFCCC).

Tetrafluoromethane ($CF_4$, PFC-14) is the simplest and most abundant PFC in the atmosphere, with a reported global mean

background mole fraction of 82.7 pmol mol$^{-1}$ (dry-air mole fraction in parts-per-trillion, ppt) in 2016 (Prinn et al., 2018). $CF_4$ has a GWP of 6500 over a 100-year time horizon (GWP$_{100}$) (Burkholder et al., 2019) and, with an estimated atmospheric lifetime of 50,000 years (Burkholder et al., 2019), it is the longest-lived greenhouse gas known. Background atmospheric mole fractions of $CF_4$ have risen sharply since the beginning of the industrial revolution. However, unlike other PFCs, which are solely anthropogenic in origin, $CF_4$ is also emitted naturally from calcium fluorites ($CaF_2$) present in continental crust

(Trudinger et al., 2016). Harnisch and Eisenhauer (1998) estimated that the fluorite reservoir is sufficient to maintain an atmospheric background mole fraction of approximately 40 ppt. More recently, this was refined to $34.05 \pm 0.33$ ppt (Trudinger et al., 2016).

The predominant anthropogenic source of $CF_4$ to the atmosphere is primary aluminium production. Kim et al. (2014) estimates that aluminium smelting accounted for ∼68% of global emissions in 2010. Emissions from the smelting of aluminium

ore occur predominantly during 'anode effects', when the feed of alumina to the cell is restricted resulting in formation of $CF_4$, but also during routine operation (Wong et al., 2015). $CF_4$ is also used commercially in the semi-conductor industry as an etchant gas for plasma etching, and as a cleaning agent in chemical vapour deposition (CVD) tool chambers. In recent years, the smelting of rare-earth metals, such as neodymium, has also been cited as a small but growing source of $CF_4$ (Cai et al., 2018; Zhang et al., 2018). Firn air measurements show that the atmospheric abundance of $CF_4$ has increased rapidly

since 1960 (Trudinger et al., 2016). While emissions peaked in 1980 and have since declined, Engel and Rigby (2019) report renewed growth in recent years.

Hexafluoroethane ($C_2F_6$, PFC-116) has an atmospheric lifetime of 10,000 years and a GWP$_{100}$ of 9200 (Burkholder et al., 2019), making it the most potent PFC, and fourth most potent greenhouse gas, listed under the Kyoto basket in terms of its GWP$_{100}$. With a global mean background mole fraction of 4.56 ppt in 2016 (Prinn et al., 2018), it is the second most abundant





PFC in the atmosphere. The presence of $C_2F_6$ has not been detected in the pre-industrial atmosphere (Trudinger et al., 2016), and its sources are very similar to the anthropogenic sources of $CF_4$, with aluminium production and the semi-conductor industry accounting for approximately one third and two thirds of the global budget in 2010, respectively (Kim et al., 2014). $C_2F_6$ is also a component of the refrigerant blend R-508 (50–70% $C_2F_6$ by weight, 30–50% trifluoromethane ($CHF_3$) by weight), which is used in very low temperature refrigeration, though this is thought to be a minor source (Kim et al., 2014).

Trudinger et al. (2016) reported peak global emissions of ~3.6 Gg yr$^{-1}$ in 2000, which was followed by a decline until 2010 and stabilization thereafter (Engel and Rigby, 2019).

Octafluoropropane ($C_3F_8$, PFC-218) has an atmospheric lifetime of 2600 years and a GWP$_{100}$ of 7000 (Burkholder et al., 2019). It is the fourth most abundant PFC, after perfluorocyclobutane (PFC-318, $c$-$C_4F_8$, see Mühle et al., 2019), with a global mean mole fraction of 0.63 ppt in 2016 (Prinn et al., 2018). The Emission Database for Global Atmospheric Research v4

(EDGAR, 2009) attributes $C_3F_8$ emissions to refrigeration/air-conditioning use and semi-conductor manufacture. While the aluminium industry is not thought to be a major source, low concentrations of $C_3F_8$ have been detected from smelter stacks (Fraser et al., 2003). At present, the aluminium industry does not account for $C_3F_8$ in their emissions reporting, and its low concentration means that it may not even be detectable by the instruments employed by the industry to monitor PFC emissions (Trudinger et al., 2016).

As a result of their exceedingly long lifetimes, PFC emissions represent a permanent (on human time-scales) alteration to the atmosphere. PFC sinks are dominated by decomposition during high temperature combustion (Cicerone, 1979; Ravishankara et al., 1993; Morris et al., 1995). Both aluminium and semi-conductor industries have targeted PFCs for emissions reductions in an effort to curb greenhouse gas emissions (Trudinger et al., 2016).

In this study, we present atmospheric PFC measurements from the Advanced Global Atmospheric Gases Experiment (AGAGE)

network. These data are used to update baseline mole fraction data and growth rates. The atmospheric measurements from European stations are used, in conjunction with an inverse modelling framework, to estimate PFC emissions for northwest Europe. We compare our emissions maps to the European Pollutant Release and Transfer Register (E-PRTR), which contains a record of PFC emissions from industrial facilities. Finally, we explore potential of using one such PFC emitting facility as a release location for a tracer experiment.

## 2   Materials and methods

### 2.1   Instrumentation

Long-term in situ PFC measurements were made by the AGAGE (Prinn et al., 2018) network. With the exception of Monte Cimone (CMN), all stations were equipped with a Medusa pre-concentration system coupled with a gas chromatograph (GC, Agilent) and quadrupole mass selective detector (MSD) (Miller et al., 2008). At CMN, measurements of $C_3F_8$ (neither $CF_4$

or $C_2F_6$ were measured) were made using an in-line auto sampler and pre-concentration device coupled to a GC mass spectrometer (Maione et al., 2013). Observations from the five 'core' AGAGE stations, Mace Head (MHD), Ireland; Trinidad Head (THD), USA; Ragged Point (RPB), Barbados; Cape Matatula (SMO), American Samoa and Cape Grim (CGO), Tas-





mania, were used to infer global trends. Observations from MHD; Jungraujoch (JFJ), Switzerland; Tacolneston (TAC), United Kingdom and CMN, Italy were used to infer northwest European emissions (Fig. 1). Station details are given in Table 1.

Each air measurement was bracketed by analysis of a working (quaternary) standard to account for short-term drifts in detector sensitivity. At JFJ and TAC, every two air measurements were bracketed. Quaternary standards were linked to a set of primary calibration scales — SIO-05 for $CF_4$ and SIO-07 for $C_2F_6$ and $C_3F_8$ — via a hierarchy of compressed real-air standards held in 34 L internally electro-polished stainless-steel canisters (Essex Industries, Missouri, USA). The estimated absolute accuracy of the calibration scales is $\sim$1.5%, $\sim$2% and $\sim$4% for $CF_4$, $C_2F_6$ and $C_3F_8$, respectively. Average measurement

precisions, estimated as the standard deviation of the bracketing standards for all sites across all years, were estimated to be $\sim$0.18%, $\sim$0.77% and $\sim$2.70% for $CF_4$, $C_2F_6$ and $C_3F_8$, respectively. Mass spectrometers were run in selective ion mode (SIM). $CF_4$ was detected using target ion $CF_3^+$ ($m/z$ 69) and qualifier ion $CF_2^{2+}$ ($m/z$ 50), $C_2F_6$ was detected using target ion $C_2F_5^+$ ($m/z$ 119) and qualifier ion $CF_3^+$ ($m/z$ 69) and $C_3F_8$ was detected using target ion $C_3F_7^+$ ($m/z$ 169) and qualifier ion $C_2F_5^+$ ($m/z$ 119). For each gas, the ratio of target to qualifying ion was continuously monitored to ensure that co-eluting species did

not interfere with the analysis. Weekly system blanks were conducted to test for system leaks and/or carrier gas impurities. MHD and TAC showed small blanks for all three PFCs. These were carefully assessed for each carrier gas cylinder. Where the blank variability was negligible, all measurements made using that cylinder were blank-corrected. Measurements coinciding with high and/or variable blanks were rejected.

## 2.2   Archived air samples

The atmospheric histories of $CF_4$, $C_2F_6$ and $C_3F_8$ were extended backwards in time via analysis of northern and southern hemispheric archived air samples (Figs. 2-4). A full description of the collection and analysis of these samples can be found in Mühle et al. (2010) and Trudinger et al. (2016). In short, northern hemispheric archive samples, which were provided by a range of laboratories, were filled under baseline conditions using a range of filling techniques and for different purposes. The southern hemisphere archive samples are part of the Cape Grim air archive (CGAA, Fraser et al. (2007)) and were cryogenically filled

into electro-polished stainless steel cylinders during baseline conditions. Northern and southern hemispheric samples were analysed at the Scripps Institution of Oceanography and the Commonwealth Scientific and Industrial Research Organisation (CSIRO), Aspendale, Australia using Medusa GCMS instruments. Each archive sample was analysed in replicate. Non-linearity data were collected prior to, during, and after each sample analysis. Blank runs were also conducted regularly, with blank corrections applied where needed.

## 2.3   Global emissions estimation

We estimate global emissions using the well-established two-dimensional AGAGE 12-box model, which simulates seasonally varying but annually repeating transport (Cunnold et al., 1983, 2002; Rigby et al., 2013). The model simulates monthly background semi-hemispheric abundances of trace-gases given the emissions from the surface into the semi-hemispheres. The model is split into four lower and upper tropospheric boxes and four stratospheric boxes. These boxes are divided zonally at 115 $\pm$30° and the equator, and vertically at 1000, 500, 200 and 0 hPa. The model performs well when the lifetime of the gases are



long compared to the inter-hemispheric exchange time, which makes it well suited to our application due to the long lifetimes of the PFCs, which is considered within the model.

Estimates of global emissions are derived using a Bayesian method in which global emission growth rates are constrained a priori (following Rigby et al., 2014). Here, we assumed that, in the absence of observations, the annual emissions growth rate
would be zero plus or minus 20% of the maximum emissions from the Emissions Database for Global Atmospheric Research v4.2 (EDGAR, 2009). The derived emissions were not found to be sensitive to reasonable changes to this constraint. The estimate is informed using monthly mean baseline-filtered observations from the five background AGAGE stations (MHD, THD, RPB, SMO and CGO), averaged into semi-hemispheric monthly means (see Rigby et al., 2013). Prior to 2004, the inversion is constrained by archived air samples from the CGAA (Fraser et al., 2007) and various archived northern hemispheric
samples (Mühle et al., 2010). Model-measurement uncertainties are assumed to be equal to the monthly baseline variability. For archived air samples, this term is assumed to be equal to the mean variability in the high-frequency baseline data, scaled by the mole fraction difference between the high-frequency mean and archived air sample. For the archived samples, the repeatability of each measurement was included in the model-measurement uncertainty. These uncertainties are propagated through the inversion, along with the uncertainty due to the prior constraint, to calculate the posterior emissions uncertainty.

## 2.4 Estimating European emissions using a regional inverse modelling technique

We infer regional PFC emissions by combining atmospheric measurements (Section 2.1) and air histories, derived using the atmospheric-dispersion model NAME (Jones et al., 2007), within the Met Office's Inversion Technique for Emissions Modelling (InTEM) framework. Numerous examples exist within the literature that describe the use of InTEM for the estimation of long-lived greenhouse gas emissions (e.g. Manning et al. (2011); Say et al. (2016); Arnold et al. (2018); Rigby et al. (2019);
Mühle et al. (2019)). In short, the simulated transport of gas in the NAME atmospheric transport model creates a sensitivity matrix that maps the spatial surface emissions to a modelled measurement. Meteorology from the UK Met Office Unified Model (UM) (Walters et al., 2019), including the nested high resolution UK model (UKV) from 2014 onwards, drives the transport through advection, diffusion and turbulence in NAME. The global UM and UKV are available at 3 and 1-hourly resolutions, respectively. The horizontal resolution of the global UM has increased from ∼40 km in 2003 to ∼12 km in 2020, the UKV is
at ∼1.5 km over the UK and Ireland. The NAME output spatial latitudinal by longitudinal resolution is 0.234° by 0.352°. We regard measurements as being sensitive to emissions from the surface when simulated particles in the model fall below 40 m above ground level (agl) within a 30-day period prior to the measurement. NAME simulates the release of 20,000 particles per hourly measurement interval within the computational domain –98 to 40°E and 11 to 79°N. At MHD and TAC, these particles were released from a 20 m vertical line centered on the sample inlet. At JFJ and CMN, the mountain meteorology presents
additional challenges for the model. At these stations, we assumed a release point of 1000 metres above ground level (m agl) for JFJ and 500 m agl for CMN, representing a compromise between the model surface altitude and the actual station height. The NAME model was run offline prior to the inversion and tailored to create a sensitivity matrix for a specific gas following Arnold et al. (2018).





InTEM estimates the spatial emissions of $CF_4$, $C_2F_6$ and $C_3F_8$ by combining measurements, the sensitivities generated using

NAME, any prior knowledge available about the emissions sources and magnitudes, and the uncertainty in these quantities. InTEM is a Bayesian inversion system that uses a non-negative least-squares solver. This is described in detail in Arnold et al. (2018). The baseline, or background, mole fraction is solved for within the inversion system. Prior baselines were estimated for MHD, JFJ and CMN using the measurements at those sites. The prior baseline for TAC was assumed to be equivalent to the prior MHD baseline, since they occupy similar latitudes.

The measurement uncertainty was estimated to be a combination of the precision described in Section 2.1 and the variability of three consecutive measurements centred around the measurement point. The observations were averaged into 4-hour windows. The model uncertainty was a combination of the prior baseline uncertainty and the magnitude of the median pollution event at the measurement location per year. Observations recorded at TAC were selected when the difference between $CH_4$ observations (made using a Picarro G2301) at different heights on the mast during the 4-hour period were less than 20 ppb, i.e.

the air was well mixed in the vertical. Analysis of the meteorology during these selected times provided thresholds that were applied at MHD which only has one measurement height. Off-diagonal elements of the model-observation uncertainty matrix were calculated by assuming a temporal correlation coefficient of 12 hours.

For each inversion, the system used an a priori estimate of emissions which were distributed uniformly over land, on which a prior distribution with very large uncertainty was placed, ensuring that the posterior solution was informed entirely by the

atmospheric measurements. Only two constraints were applied a priori: 1) All grid cells were forced to have a non-negative emission; 2) the 9 grid cells centred on the locations of known PFC emitters, as reported to the E-PRTR, were specifically solved for. The E-PRTR only reports total PFC emissions (e.g. the sum total of all individual PFCs). Therefore, we define all E-PRTR locations for all three PFCs. Elsewhere in the domain, the spatial resolution of the underlying grid was allowed to vary within each country, with finer resolution in areas found to have high emissions and high sensitivity to the measurements.

## 3    Results

### 3.1    Atmospheric trends and global emissions

The modelled and measured baseline atmospheric trends (1979 - 2019) of $CF_4$, $C_2F_6$ and $C_3F_8$ are shown in Figs. 2 - 4. Prior to the calculation of monthly baseline estimates, the AGAGE pollution algorithm (Cunnold et al., 2002) was used to remove regional pollution effects observed at each station.

For $CF_4$, a large increase in baseline mole fraction is evident across all semi-hemispheres. The global mean increased from 52.1 ppt in 1979 to 85.5 in 2019, representing a 64% increase. We observed significant variations in growth rate across the measurement window. In 1979, the growth rate was estimated to be 1.3 ppt yr$^{-1}$, but declined steadily to a minimum of 0.6 ppt yr$^{-1}$ by 2009. The drop in growth rate observed around 2009 is probably due to a fall in aluminium production following the 2008 financial crisis. Production of primary aluminium dropped by roughly 6% ($\sim$ 200,000 metric tons) between 2008

and 2009 (IAI, 2020). Global emissions declined by 0.6 Gg over the same period (Table 2). However, more recent years have seen a renewed increase in the growth rate of $CF_4$, rising from 0.7 ppt yr$^{-1}$ in 2010 to 0.9 ppt yr$^{-1}$ in 2018. At 14.1 Gg yr$^{-1}$,





emissions in 2018 are the largest observed since high-frequency measurements began. Our work is consistent with Trudinger et al. (2016) and Engel and Rigby (2019) who showed the first increase in global $CF_4$ emissions since the 1980s. Despite this, given global primary aluminium production has increased $\sim$5-fold over the last 40 years (IAI, 2020), our estimates highlight the success of efficiency improvements and abatement technology in reducing emissions of $CF_4$ from the industry.

The global $C_2F_6$ baseline mole fraction grew from 1.1 ppt in 1979 to 4.8 ppt in 2019, an increase of more than 4-fold. The relative increase is substantially larger than that of $CF_4$, suggesting that there are major sources of $C_2F_6$ not linked to the aluminium industry. The growth rate peaked in 1999 at an estimated 0.13 ppt $yr^{-1}$, followed by a sustained period of decline to a minimum of 0.07 ppt $yr^{-1}$ in 2009. As with $CF_4$, the minimum rate of growth in 2009 is probably a result, at least in part, of the reduced demand for aluminium following the 2008 financial crisis. The effect of the crisis on demand for electronics, and therefore the consumption of $C_2F_6$ as an etchant gas, is not known. The 2009 minimum was followed by a period of stagnation, with a near constant growth rate between 2009 and 2013. Annual global emissions did not vary significantly during this period, remaining stable at $\sim$1.9 Gg $yr^{-1}$. However, there is some evidence for a resurgence in $C_2F_6$ emissions post-2013. The global growth rate in 2017 was estimated to be 0.09 ppt $yr^{-1}$, with corresponding global emissions of 2.3 Gg $yr^{-1}$, the largest observed since prior to the financial crisis.

The global $C_3F_8$ baseline mole fraction grew from an estimated 0.07 ppt in 1983 to 0.68 ppt in 2019. The large almost 10-fold increase shows that, in terms of relative growth, emissions of $C_3F_8$ have increased sharply when compared to $CF_4$ and $C_2F_6$. Unlike $CF_4$ and $C_2F_6$, the aluminium industry is not a major contributor of global emissions of $C_3F_8$, though detectable concentrations have been observed in the outflow from smelter stacks (Fraser et al., 2003). Following a period of relative stability between 1985–1992, the growth rate increased rapidly, reaching a maximum of 0.03 ppt $yr^{-1}$ in 2003. Thereafter, a period of steady decline saw the growth rate fall to 0.015 ppt $yr^{-1}$ in 2014, with corresponding global emissions of 0.51 (0.48 – 0.55) Gg $yr^{-1}$. Of the three PFCs discussed, $C_3F_8$ is the only gas for which a pronounced 'dip' in growth rate was not observed around the time of the financial crisis, perhaps indicative of the resilience of the semi-conductor industry to the crisis relative to aluminium producers. Since 2015, the global growth rate has remained comparatively stable, with no statistically significant trend in global emissions. Emissions in 2019 were estimated to be 0.56 (0.51 – 0.60) Gg $yr^{-1}$.

### 3.2 Northwest European PFC emissions

We present estimates of PFC emissions in northwest Europe (the United Kingdom; Ireland; France; Belgium, the Netherlands and Luxembourg (collectively termed Benelux); and Germany) using the procedure described in Section 2.4, based on the sensitivity of the measurements to emissions from these countries (see Fig. 1).

From 2005 – 2007 (2005 – 2010 for $CF_4$), we only report emissions for the UK, Ireland and Benelux, owing to the lack of atmospheric measurements from continental Europe and therefore sensitivity to southern France and eastern Germany during this period. Reported estimates for France and Germany (and the northwest Europe total) begin in 2008 ($C_2F_6$ and $C_3F_8$) and 2010 ($CF_4$), corresponding with the availability of measurements from JFJ (Table 1).





### 3.2.1 CF$_4$

Annual CF$_4$ emissions for northwest Europe are shown in Fig. 5 and Table S1. Owing to the considerable uncertainties, we find no statistically significant trend in emissions from the region over the measurement period.

European aluminium production dropped significantly during the measurement period, declining from 3.2 MT in 2004, to just 2.2 MT in 2016 (IAI, 2020). In that time the number of active European aluminium smelters also declined, falling from 25 to 16. On average, northwest Europe accounted for 2.1% of global emissions in 2010 (0.21 Gg in northwest Europe; 10.2

220 Gg globally), but only 0.7% in 2018 (0.1 Gg in northwest Europe; 14.1 Gg globally). Despite no significant trend in northwest Europe, global emissions increased considerably over the same period (Table 2), indicating that emissions from other regions have increased. China was the largest producer of primary aluminium in 2019 (IAI, 2020). Comparison of our northwest European estimates with compiled emissions reported to the UNFCCC indicates a discrepancy between reporting methods (Fig. 5) throughout most of the reporting window. Our estimates are typically larger than the inventory, with better agreement

after 2015.

Of the individual countries/regions examined, Ireland is the only country whose reported emissions to the UNFCCC have increased in recent years. The InTEM estimates mirror this trend, though the uncertainties are considerable. Our estimates increased from 2.2 (0.0 – 5.7) Mg yr$^{-1}$ in 2012 to 10.6 (4.3 – 16.9) Mg yr$^{-1}$ in 2019. Ireland is not a producer of aluminium metal, though it does have a bauxite refinery. Therefore, its emissions of CF$_4$ are probably due to consumption by semi-

230 conductor manufacturers. Across all years of the E-PRTR (2007 – 2017), Ireland only reported emissions from a single facility, a semi-conductor factory to the west of Dublin. In the early and later parts of the record, this source is evident in our spatial maps, but it is not seen in between 2010 and 2016. It is therefore probable that emissions from this site are the main driver of trends in Ireland's CF$_4$ emissions.

The inferred spatial distribution of northwest European CF$_4$ emissions is shown in Fig. 6. In 2005, regional emissions

were dominated by those from west Germany, roughly consistent (accounting for transport errors) with the location of three aluminium smelters in the Ruhr valley. Aluminium production is also the probable source of smaller emissions from southwest Norway and northern Denmark, although we do not report national estimates for these countries. Aluminium production in the Ruhr valley in Germany remains a significant source of CF$_4$ in later years, though emissions from other areas and industries become apparent. Starting in 2010, strong emissions were found for southeast France and northwest Italy. Southeast France has

previously been linked to emissions of other halogenated species (Maione et al., 2014). Our results show that emissions from this region diminished after 2015. In contrast, emissions from northwest Italy continued to grow until the end of the record. The E-PRTR lists two PFC emitters in the region. The largest of these sources, a chemicals manufacturer located near to the Italian city of Alessandria, is consistent with our emissions maps after 2010. This manufacturer reported total PFC emissions of 185 Mg in 2012, making it one of the largest emitters in Europe. Interestingly, this region has previously been linked to considerable

emissions of the hydrofluorocarbon, trifluoromethane (CHF$_3$, HFC-23) (Keller et al., 2011), though we are unable to ascertain whether theses gases share a common source.





In comparison to other large European countries, emissions from the UK and Ireland are small throughout the reporting window. Between 2005 and 2012, emissions from northern England are consistent with the location of the Lynemouth aluminium smelter, which was closed in March 2012. After this date, the predominant UK source is located near to the city of Manchester,
consistent with the location of an electronics manufacturer.

Emissions from the Benelux region are small and the uncertainties are large. In later years, the majority of these emissions are consistent with the location of chemical manufacturers. These emissions are in a similar location to a strong source of $c$-$C_4F_8$ reported by Mühle et al. (2019), which the authors attributed to the consumption of $c$-$C_4F_8$ as an intermediate feed stock in the manufacture of polytetrafluoroetheylene (PTFE). There is no available evidence supporting the use of $CF_4$ in PTFE
manufacture, though the production of $CF_4$, perhaps as a by-product, cannot be ruled out.

### 3.2.2   $C_2F_6$

Annual $C_2F_6$ emissions are shown in Fig. 7 and Table S2. Like $CF_4$, the uncertainty of our estimates for Germany and France, particularly in early years, is large and overall, there is no statistically significant trend in emissions from northwest European over the measurement period. When compared to global emissions, the average contribution of northwest Europe declined
from 2.4% in 2008 (0.05 Gg in northwest Europe; 2.1 Gg globally) to 1.6% in 2018 (0.03 Gg in northwest Europe; 2.2 Gg globally).

Our work is in reasonable agreement with $C_2F_6$ emissions reported to the UNFCCC, particularly for Ireland between 2006 and 2011, where our top-down estimates captured the significant fall in emissions reported in Ireland's national inventory. Elsewhere, our uncertainties typically overlap the average UNFCCC estimates, with the exception of Benelux, where our
emissions for 2013 – 2016 are significantly larger than the inventory, suggesting an under-reporting of emissions during this period.

$C_2F_6$ emissions maps are shown in Fig. 8. Unlike $CF_4$, whose emissions appear to be dominated by the aluminium industry, the $C_2F_6$ distribution is consistent with a greater contribution from the electronics industry (Kim et al., 2014). For instance in 2005, the two largest sources (on the outskirts of Dublin, Ireland and Paris, France) correspond with the locations of electronics
manufacturers, suggesting these emissions are linked with the consumption of $C_2F_6$ as an etching gas. In contrast, emissions from the Ruhr valley in Germany are small. By 2012, a source in northern Belgium, which is concurrent with the location of a basic chemicals manufacturer, dominates European emissions. The manufacture of basic chemicals also appear to be a source of $C_2F_6$ in southern France and northwest England. In contrast, the emissions 'hot-spot' located near to Dublin in 2005 is greatly diminished by 2012, in line with the planned phase-out of $C_2F_6$ in favour of $NF_3$, by the manufacturer.

By 2019, the spatial distribution of emissions is more varied. Large emissions are found for the Ruhr valley in Germany and likely originate from three aluminium smelters in the region that were also found to emit $CF_4$, which is known to be co-emitted during the smelting process at a ratio of around 0.1 kg/kg $CF_4$/$C_2F_6$ (Kim et al., 2014). In Ireland, emissions associated with electronics manufacture, located on the outskirts of Dublin, have ceased. However, significant emissions are now found further to the southwest. This source region does not appear to be listed under the E-PRTR, and may be a contributing factor in
the small discrepancy between our work and the UNFCCC, from 2011 onwards. These emissions are situated in a comparable



location to the Irish city of Limerick. While several electronics manufacturers are based here, these sources cannot be confirmed without further information from individual companies.

### 3.2.3 $C_3F_8$

Northwest European $C_3F_8$ emissions are shown in Fig. 9 and Table S3. As with $CF_4$ and $C_2F_6$, northwest European emissions of $C_3F_8$ exhibited no statistically significant trend over the measurements period. The contribution of northwest Europe to global emissions is considerably greater than other PFCs – 4.8% in 2008 (0.03 Gg in northwest Europe; 0.69 Gg globally) and 5.1% by 2018 (0.03 Gg in northwest Europe; 0.55 Gg globally). $C_3F_8$ is the only PFC for which northwest Europe's emissions increased relative to the global total. Several countries, including France and Ireland, reported no emissions of $C_3F_8$ to the UNFCCC across the measurement window. In general, our work is in agreement with these reports, with the uncertainty bounds of our estimates typically encapsulating 0 Mg yr$^{-1}$. Our work shows the UK to be the largest emitter of $C_3F_8$ in northwest Europe. In the early years of the record, and again after 2016, there is a significant discrepancy between reporting methods, potentially indicative of under-reporting by UK emitters.

The spatial maps reveal few notable sources of $C_3F_8$ across continental Europe. The apparent reduction of continental sources in later years may be due to the substitution of $C_3F_8$ in the semiconductor industry by lower GWP alternatives. However, it is also likely that in the absence of European measurements prior to 2008 (when JFJ came online), the inversion has less skill in inferring point source emissions at such a distance from the receptor (MHD). In 2012, the spatial maximum in emissions from the Benelux region are consistent with that found for $C_2F_6$ and $c$-$C_4F_8$ (Mühle et al., 2019), probably due to close vicinity of chemical industries, or perhaps due to a common PFC source.

UK emissions of $C_3F_8$ appear to be dominated by a single source located in northwest England (Fig. 10). Facility listings from the E-PRTR show a PFC manufacturer whose location is consistent with this source (Fig. 10). This company is the only known manufacturer of $C_3F_8$ in the UK, and possibly in the whole of Europe.

### 3.2.4 UK $C_3F_8$ emissions as a tracer for atmospheric transport

If the facility in northwest England is the only source of $C_3F_8$ in northwest Europe, this gas could potentially be used as a tracer species for the validation of atmospheric transport models, assuming that emissions are well defined. To test the validity of UK $C_3F_8$ emissions as a potential tracer, the mole fraction at MHD was modelled by multiplying the NAME sensitivity matrices (see Section 2.4) with an emissions grid, where the UK's total reported emissions of $C_3F_8$ in 2014 (UNFCCC, 8.69 Mg yr$^{-1}$) were placed in the grid cell corresponding with the location of the PFC manufacturer. The resulting time-series was then compared to the MHD observations. Fig. 11 shows the comparison of the forward model with MHD $C_3F_8$ observations for July - August 2014 (time-frame chosen for illustrative purposes). In general, pollution events observed at MHD coincide with modelled events, though in the example shown, the magnitude of the modelled events is significantly smaller than the observed mole fraction. This indicates that the assumed emissions rate is too small, or there are errors in the transport model (NAME). Alternatively, the current assumption, that emissions from the site are released at a constant rate throughout the year, may also be an over-simplification – emissions are likely to vary depending on the rate of production.



While atmospheric dispersion models, such as NAME, are used extensively to simulate atmospheric transport, they rely on
simulated atmospheric dynamics that are subject to considerable uncertainties. If emissions from this source in northwest England were known better (e.g. by collaboration with the company), then differences between the forward model and observed mole fractions might be used to improve model transport or make estimates of transport uncertainties. Due to the long atmospheric lifetime of $C_3F_8$ photo-chemical loss processes do not need to be taken into account, thus further reducing uncertainty. Quantifying these differences would be a useful means by which to assess and perhaps improve the performance of individual
model simulations in future inverse modelling work.

## 4   Conclusions

We have presented measurements of tetrafluoromethane ($CF_4$, PFC-14), hexafluoroethane ($C_2F_6$, PFC-116) and octafluoropropane ($C_3F_8$, PFC-218) from the AGAGE network. We combined measurements from five background stations, in conjunction a box model, to infer global trends. For $CF_4$, the global mean baseline mole fraction increased by ~33 ppt between 1979
and 2019. The global growth rate declined across much of the measurement period, falling to a minimum of 0.6 ppt $yr^{-1}$ around the time of 2008 financial crisis. However, the growth rate began to rise again after 2011, consistent with increasing global emissions. For $C_2F_6$, a maximum growth rate of 0.125 ppt $yr^{-1}$ was observed around 1999, followed by a period of steady decline. Like $CF_4$, we found a renewed increase in the global growth rate after 2011. $C_3F_8$ exhibited a rapid increase in growth rate, starting in the early 1990s and ending in the early 2000s, followed by steady decline until 2013. In recent years, a
small increase in growth rate was observed.

We used observations from four European observatories to infer PFC emissions from northwest Europe. Between 2010 and 2019, northwest European emissions of $CF_4$ exhibited no statistically significant trend, despite an increase in global aluminium production and continued demand for electronic components, consistent with growth in emissions from other regions. In early years, emissions were predominantly consistent with the locations of aluminium smelters. However, in more recent years the
largest source of $CF_4$ was northwest Italy These emissions might be linked to a chemicals manufacturer that reports substantial PFC emissions to the E-PRTR.

Likewise for $C_2F_6$, no significant trend was observed for northwest European emissions from 2008 until 2019. A notable fall in emissions from Ireland was observed between 2007 and 2012. This trend was mirrored by Ireland's national inventory report, and appears to be linked with an electronics manufacture on the outskirts Dublin. By 2019, the largest source of $C_2F_6$
in northwest Europe was West Germany, most notably the Ruhr valley region in which three aluminium smelters are found. Northwest European $C_3F_8$ emissions were stable over the measurement period. Several countries, including France and Ireland, reported no emissions of $C_3F_8$, which our results are consistent with. With the exception of a small source in Benelux, we found northwest European emissions of $C_3F_8$ were dominated by a single source in northwest England, consistent with the location of a PFC manufacturer. We explored the potential of using this facility in a tracer release experiment and showed that, if high
frequency data were made available, emissions from this site could be used to provide useful information related to transport model performance.





*Code and data availability.*  Atmospheric measurement data from AGAGE stations are available from the AGAGE website (http://agage.mit.edu/data/agage-data). Data from the Tacolneston observatory are available from the Centre for Environmental Data Analysis (CEDA) data archive (https://catalogue.ceda.ac). AGAGE 12-box model code will be made available upon request by contacting Matt Rigby. Licences to use NAME and InTEM are available
for research purposes via a request to the UK Met Office or on request from Alistair Manning.

*Author contributions.*  Measurement data were collected by SOD, DY, AW, DS, SR, MKV, MM, JA, PBK, JM RFW and RGP. CMH produced and maintained the gravimetric SIO calibration scales for these gases. AJM conducted the NAME runs and ran the InTEM inverse model. MR performed the global 12-box model inversions. BE performed the $C_3F_8$ forward model runs. DS and LW analysed InTEM and forward model output. DS and LW wrote the manuscript, with contributions from all co-authors.

*Competing interests.*  The authors declare that they have no competing interests.

*Acknowledgements.*  The authors would like to thank the technicians for their diligent maintenance of the instruments at each measurement site. The operations of Mace Head and Tacolneston were funded by the UK Department of Business, Energy and Industrial Strategy (BEIS) through contract 1537/06/2018 to the University of Bristol. The operations of Mace Head and Ragged Point were also partly funded under NASA contract NNX16AC98G to MIT with a sub-award 5710002970 to the University of Bristol. Ragged Point was also partly funded by
NOAA grant RA133R15CN0008 to University of Bristol. Support for the observations at Jungfraujoch comes through the Swiss National Programs HALCLIM and CLIMGAS-CH (Swiss Federal Office for the Environment, FOEN), by the International Foundation High Altitude Research Stations Jungfraujoch and Gornergrat (HFSJG) and by ICOS-CH (Integrated Carbon Observation System Research Infrastructure). Observations at Cape Grim are supported largely by the Australian Bureau of Meteorology, CSIRO, and NASA contract NNX16AC98G to MIT with sub-award 5710004055 to CSIRO. Operations at the 'O.Vittori' station (Monte Cimone) are supported by the National Research
Council of Italy. Trinidad Head, Cape Matatula and data processing and calibration across the AGAGE network were funded by NASA grants NNX16AC96G and NNX16AC97G to the Scripps Institution of Oceanography.



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





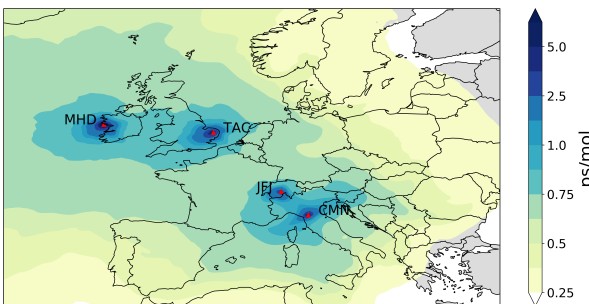

**Figure 1.** Average footprint emission sensitivity in picoseconds per mole (ps/mol) obtained from NAME 30-day backward calculations for the four measurement sites Mace Head (MHD), Tacolneston (TAC), Jungfraujoch (JFJ) and Mt. Cimone (CMN) over 2015. Measurement sites are marked as red triangles.





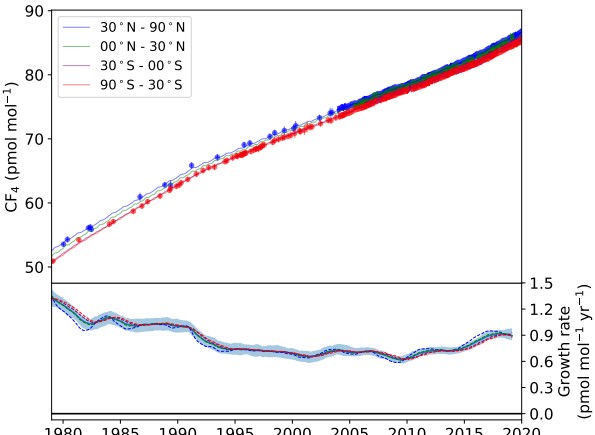

**Figure 2.** Modelled semi-hemispheric monthly average CF$_4$ mole fractions (30–90 N: blue; 0–30 N: green; 30–0 S: purple; 90–30 S: red). Averaged observations are shown as data points with $1\sigma$ error bars). The more sparse filled circles represent northern (blue) and southern (red) hemispheric air archive samples. The solid trend lines were calculated using the AGAGE 12-box model with emissions from the inversion as input. The lower plot shows the model derived mole fraction growth rate, smoothed with an approximate 1-year filter, for each semi-hemisphere and the global mean with $1\sigma$ uncertainty (solid line and shading).





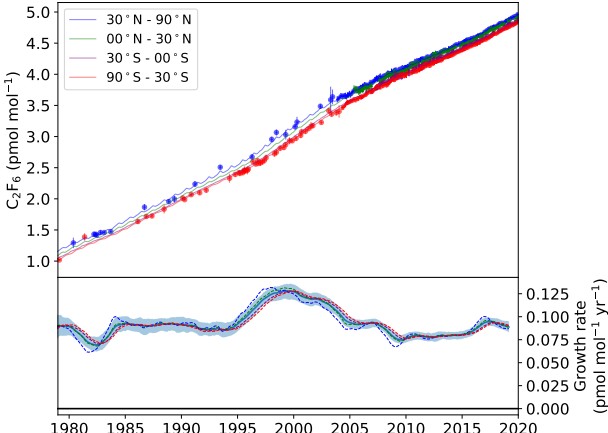

**Figure 3.** Modelled semi-hemispheric monthly average $C_2F_6$ mole fractions (30–90 N: blue; 0–30 N: green; 30–0 S: purple; 90–30 S: red). Averaged observations are shown as data points with $1\sigma$ error bars). The more sparse filled circles represent northern (blue) and southern (red) hemispheric air archive samples. The solid trend lines were calculated using the AGAGE 12-box model with emissions from the inversion as input. The lower plot shows the model derived mole fraction growth rate, smoothed with an approximate 1-year filter, for each semi-hemisphere and the global mean with $1\sigma$ uncertainty (solid line and shading).





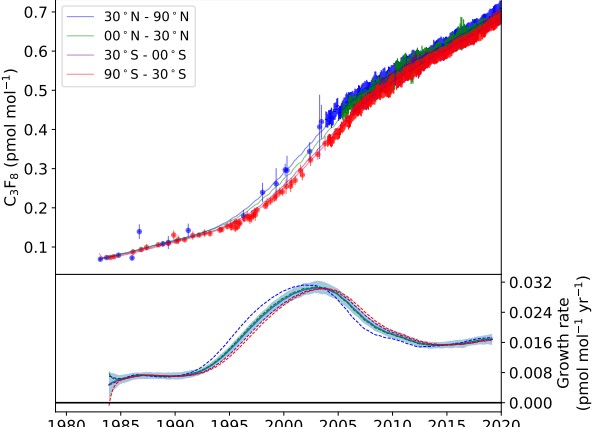

**Figure 4.** Modelled semi-hemispheric monthly average $C_3F_8$ mole fractions (30–90 N: blue; 0–30 N: green; 30–0 S: purple; 90–30 S: red). Averaged observations are shown as data points with $1\sigma$ error bars). The more sparse filled circles represent northern (blue) and southern (red) hemispheric air archive samples. The solid trend lines were calculated using the AGAGE 12-box model with emissions from the inversion as input. The lower plot shows the model derived mole fraction growth rate, smoothed with an approximate 1-year filter, for each semi-hemisphere and the global mean with $1\sigma$ uncertainty (solid line and shading).





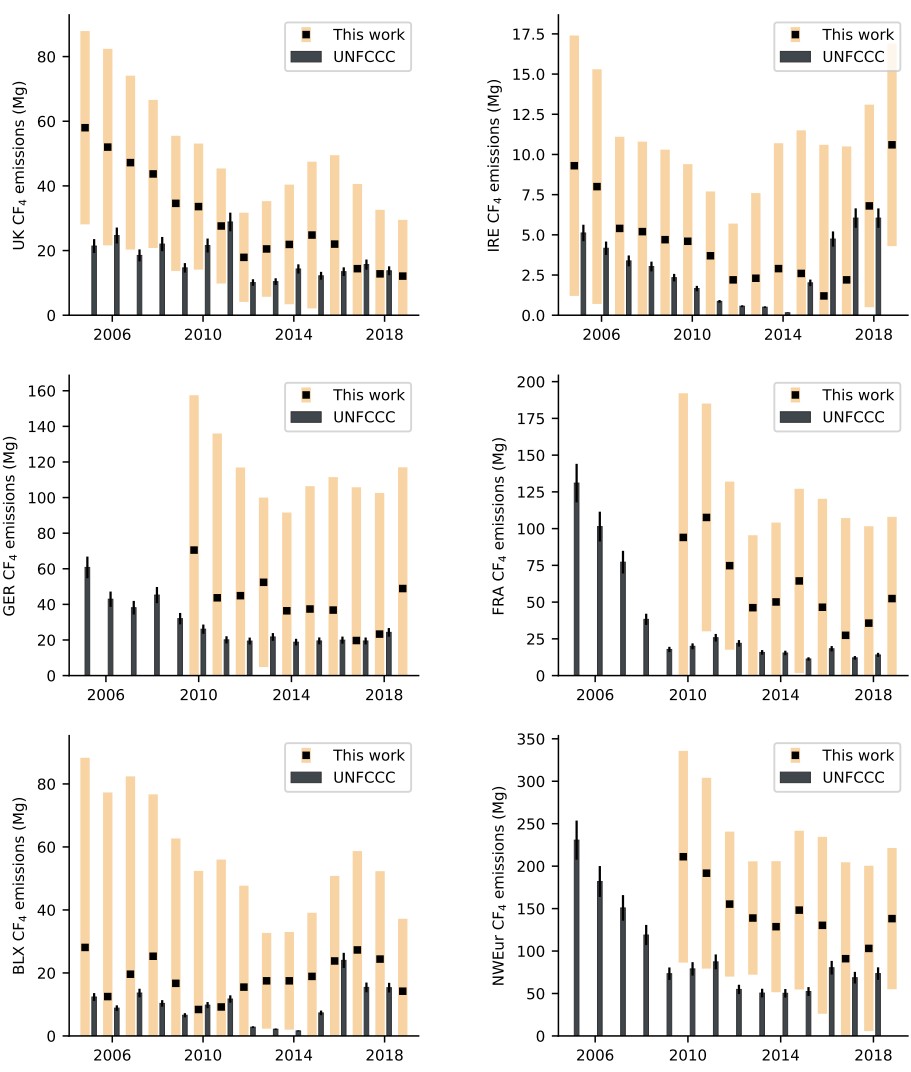

**Figure 5.** Annual $CF_4$ emissions (2005 – 2019) for northwest European countries, in Mg yr$^{-1}$. InTEM estimates are shown as black squares with pale orange uncertainty bounds. Emissions reported to the UNFCCC (sum of individual reporting countries) are shown as black bars, with an assumed uncertainty (black error bars) of 10%. Note that UNFCCC data are only available up until 2018 inclusive.



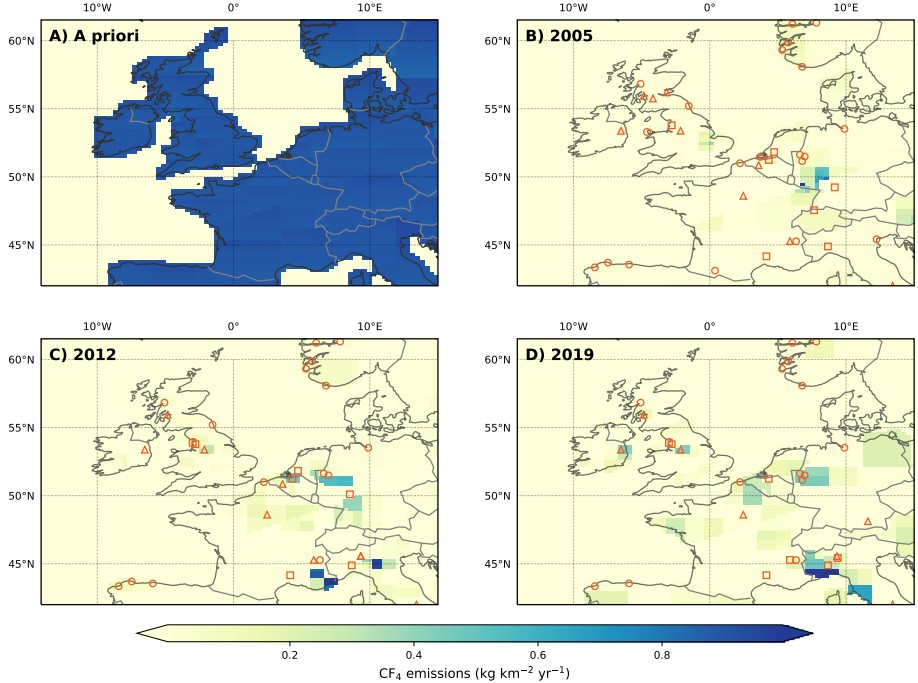

**Figure 6.** Northwest European $CF_4$ emissions, in kg km$^{-2}$ yr$^{-1}$. A) The a priori emissions field. With the exception of the oceans, emissions were distributed uniformly across the model domain. A posteriori emissions are shown for B) 2005, C) 2012 and D) 2019. Facilities that reported PFC emissions to the E-PRTR in the selected year are shown as orange circles (aluminium smelters), triangles (electronics manufacturers) and squares (chemical manufacturers, including petroleum products). Since the reporting period for the E-PRTR is shorter than that of our measurements, 2005 and 2019 emissions are compared to the earliest (2007) and latest (2017) years of the E-PRTR database, respectively. Note that the E-PRTR database reports cumulative PFC emissions.





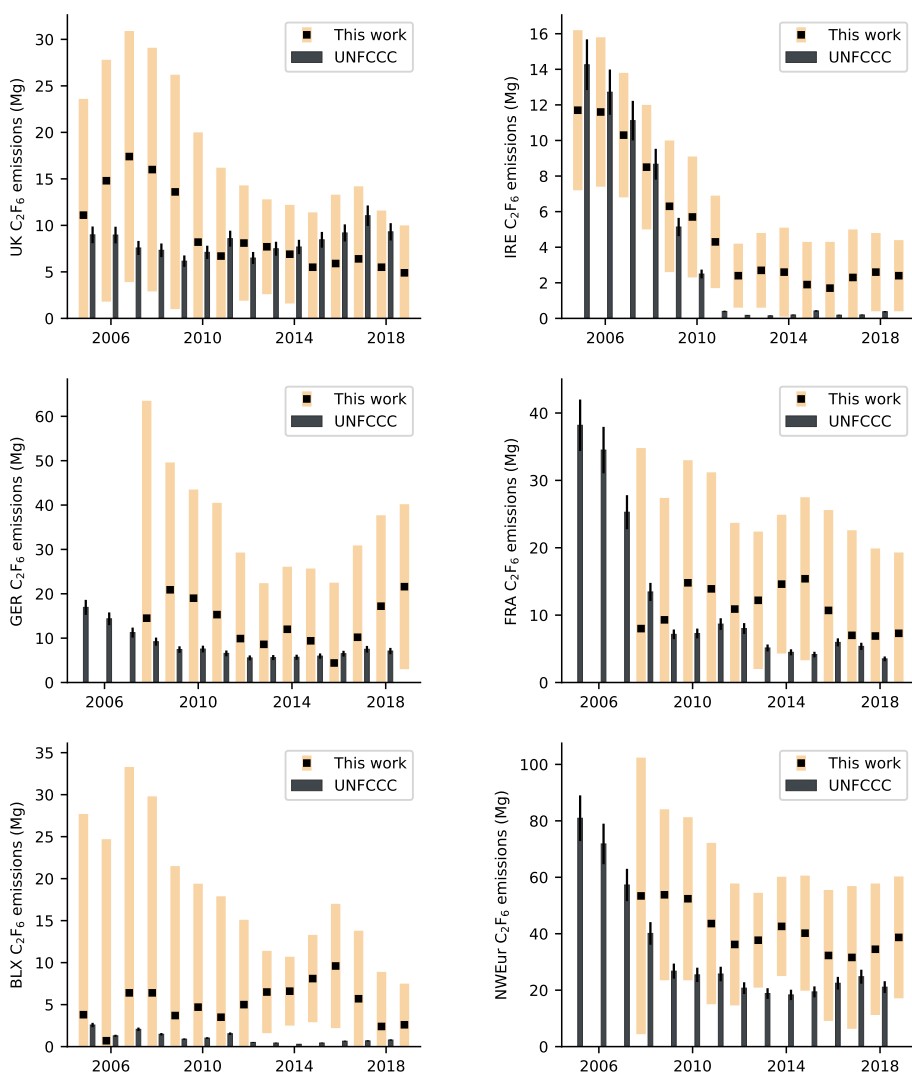

**Figure 7.** Annual C$_2$F$_6$ emissions (2005 – 2019) for northwest European countries, in Mg yr$^{-1}$. InTEM estimates are shown as black squares with pale orange uncertainty bounds. Emissions reported to the UNFCCC (sum of individual reporting countries) are shown as black bars, with an assumed uncertainty (black error bars) of 10%. Note that UNFCCC data are only available up until 2018 inclusive.

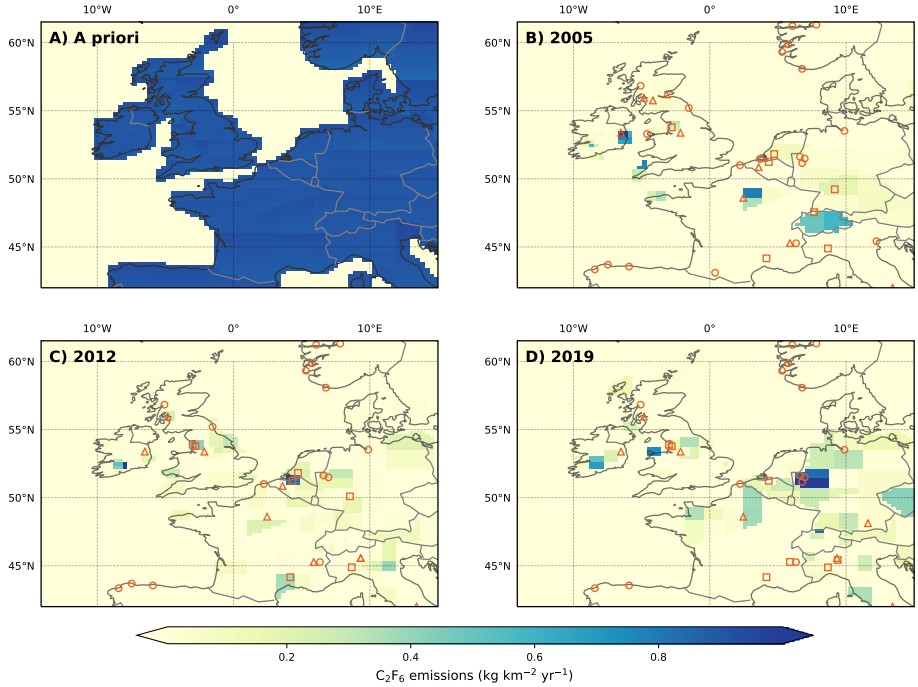

**Figure 8.** Northwest European $C_2F_6$ emissions, in kg km$^{-2}$ yr$^{-1}$. A) The a priori emissions field. With the exception of the oceans, emissions were distributed uniformly across the model domain. A posteriori emissions are shown for B) 2005, C) 2012 and D) 2019. Facilities that reported PFC emissions to the E-PRTR in the selected year are shown as orange circles (aluminium smelters), triangles (electronics manufacturers) and squares (chemical manufacturers, including petroleum products). Since the reporting period for the E-PRTR is shorter than that of our measurements, 2005 and 2019 emissions are compared to the earliest (2007) and latest (2017) years of the E-PRTR database, respectively. Note that the E-PRTR database reports cumulative PFC emissions.



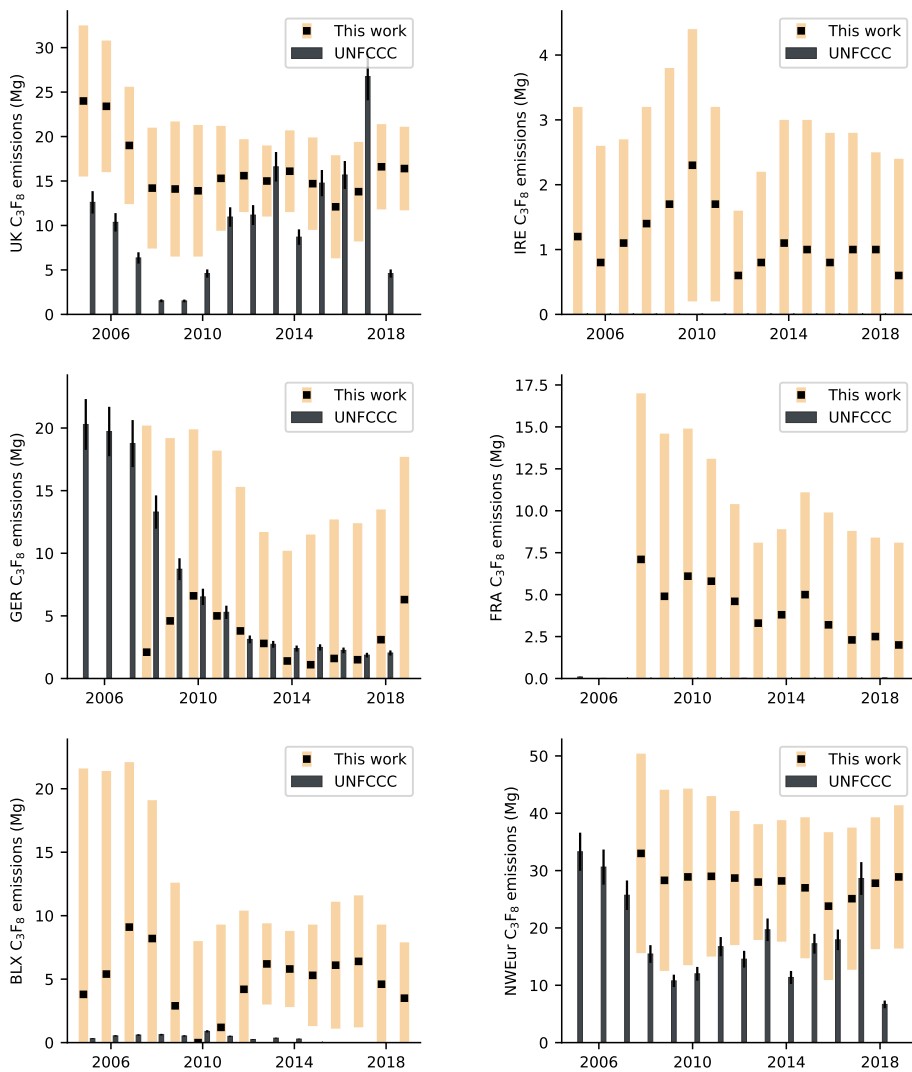

**Figure 9.** Annual $C_3F_8$ emissions (2005 – 2019) for northwest European countries, in Mg yr$^{-1}$. InTEM estimates are shown as black squares with pale orange uncertainty bounds. Emissions reported to the UNFCCC (sum of individual reporting countries) are shown as black bars, with an assumed uncertainty (black error bars) of 10%. Note that UNFCCC data are only available up until 2018 inclusive.





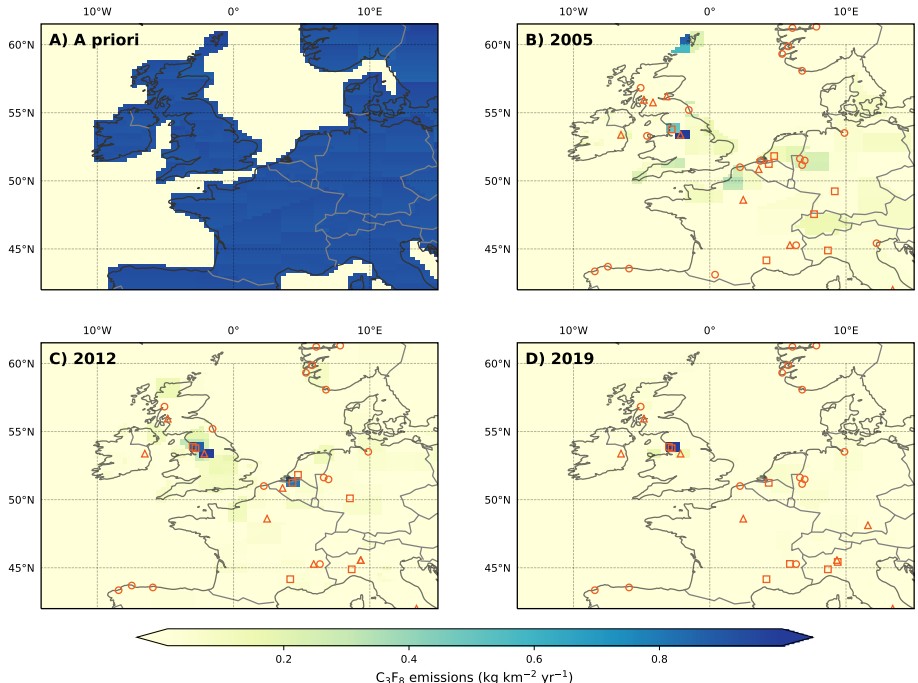

**Figure 10.** Northwest European $C_3F_8$ emissions, in kg km$^{-2}$ yr$^{-1}$. A) The a priori emissions field. With the exception of the oceans, emissions were distributed uniformly across the model domain. A posteriori emissions are shown for B) 2005, C) 2012 and D) 2019. Facilities that reported PFC emissions to the E-PRTR in the selected year are shown as orange circles (aluminium smelters), triangles (electronics manufacturers) and squares (chemical manufacturers, including petroleum products). Since the reporting period for the E-PRTR is shorter than that of our measurements, 2005 and 2019 emissions are compared to the earliest (2007) and latest (2017) years of the E-PRTR database, respectively. Note that the E-PRTR database reports cumulative PFC emissions.





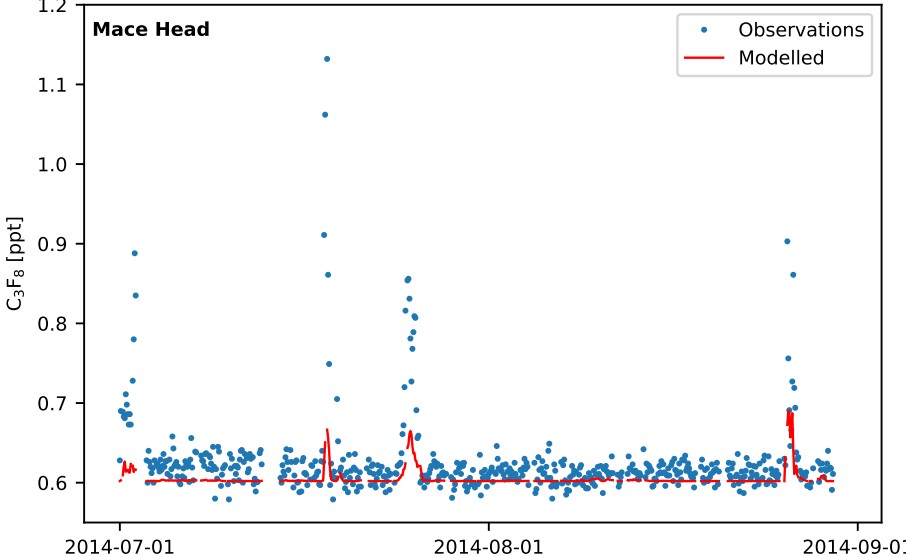

**Figure 11.** Comparison of measured (blue dots) and modelled (red line) $C_3F_8$ mole fractions at MHD for Jul - Aug 2014. Dates are chosen for illustrative purposes only.



**Table 1.** Station names, locations and inlet heights for the instruments used to quantify global and regional PFC emissions. Note that for the mountain site, JFJ, the inlet is situated below the instrument.

| Site name | Acronym | Location | Altitude (m asl) | Inlet (m agl) | Data range |
|---|---|---|---|---|---|
| Cape Grim | CGO | 41° S, 145° E | 94 | 10 | Jan 2004 – present |
| Cape Matatula | SMO | 14° S, 171° W | 82 | 10 | May 2006 – present |
| Jungfraujoch | JFJ | 47° N, 8° E | 3580 | -15[a] | Apr 2008 – present |
| Mace Head | MHD | 53° N, 10° W | 8 | 10 | Dec 2003 – present |
| Monte Cimone | CMN | 44° N, 11° E | 2165 | 10 | Jan 2008 – present |
| Ragged Point | RPB | 13° N, 59° W | 32 | 10 | May 2005 – present |
| Tacolneston | TAC | 53° N, 1° E | 56 | 185[b] | Aug 2012 – present |
| Trinidad Head | THD | 41° N, 124° W | 107 | 10 | Apr 2005 – present |

[a] The original JFJ inlet was situated at 10 m agl. The instrument began sampling from the -15 m agl inlet on the $15^{th}$ August 2012.

[b] The original TAC inlet was situated at 100 m agl. The instrument began sampling from the 185 m agl inlet on the $10^{th}$ March 2017.





**Table 2.** Global annual PFC emissions, estimated using the AGAGE 12-box model (an extension of the work by Rigby et al. (2014), Trudinger et al. (2016) and Engel and Rigby (2019), in gigagrams per year (Gg yr$^{-1}$). Lower and upper uncertainty bounds correspond to the $16^{th}$ and $84^{th}$ percentiles of the posterior model distribution, respectively.

|      | $CF_4$ | $C_2F_6$ | $C_3F_8$ |
| --- | --- | --- | --- |
| 2005 | 10.9 (10.0 – 12.0) | 2.3 (2.2 – 2.5) | 0.93 (0.87 – 0.98) |
| 2006 | 11.1 (10.5 – 12.2) | 2.3 (2.2 – 2.4) | 0.85 (0.80 – 0.89) |
| 2007 | 10.9 (10.1 – 11.6) | 2.3 (2.2 – 2.4) | 0.76 (0.72 – 0.80) |
| 2008 | 10.3 (9.5 – 11.2) | 2.1 (1.9 – 2.2) | 0.69 (0.65 – 0.73) |
| 2009 | 9.7 (8.9 – 10.5) | 1.9 (1.7 – 2.0) | 0.64 (0.59 – 0.68) |
| 2010 | 10.2 (9.4 – 11.0) | 1.9 (1.8 – 2.1) | 0.61 (0.56 – 0.65) |
| 2011 | 10.9 (9.8 – 11.5) | 1.9 (1.8 – 2.1) | 0.57 (0.53 – 0.60) |
| 2012 | 11.2 (10.3 – 11.9) | 1.9 (1.8 – 2.0) | 0.53 (0.49 – 0.58) |
| 2013 | 11.2 (10.3 – 12.1) | 1.9 (1.8 – 2.1) | 0.52 (0.48 – 0.56) |
| 2014 | 11.3 (10.4 – 12.1) | 2.0 (1.8 – 2.1) | 0.51 (0.48 – 0.55) |
| 2015 | 12.1 (11.1 – 13.0) | 2.0 (1.8 – 2.1) | 0.52 (0.48 – 0.55) |
| 2016 | 13.0 (12.2 – 14.0) | 2.1 (2.0 – 2.3) | 0.52 (0.49 – 0.55) |
| 2017 | 13.9 (13.0 – 15.0) | 2.3 (2.1 – 2.4) | 0.54 (0.50 – 0.57) |
| 2018 | 14.1 (13.2 – 15.1) | 2.2 (2.1 – 2.4) | 0.55 (0.50 – 0.59) |
| 2019 | 13.9 (12.8 – 15.4) | 2.2 (2.0 – 2.4) | 0.56 (0.51 – 0.60) |