# Peer review of "Global trends and European emissions of tetrafluoromethane (CF4), hexafluoroethane (C2F6) and octafluoropropane (C3F8)"

_Atmospheric Chemistry and Physics, 2020_

## Referee Comment (RC1) · Anonymous Referee #1 · 11 Nov 2020

This paper uses atmospheric measurements from the AGAGE network to infer global trends and northwest European emissions of 3 PFCs (CF4, C2F6 and C3F8). The paper is very well written and makes an important contribution. The work on global trends updates previous studies of PFCs using the AGAGE network, but it is important to continue to monitor the trends. The work on European emissions is new and very interesting. I have only minor comments.

line 13-14 "Unlike CF4 (and to a lesser extent C2F6), we observed no clear minimum associated with the 2008 financial crisis for C3F8." This sentence could be expressed more clearly to state what was observed for each of the 3 PFCs. It is currently written

[Figure]

in a bit of a backwards way.

Fig 2-4 I assume the symbols are monthly averages of baseline measurements, should "baseline" be mentioned in the figure captions?

line 182 - add "in 2002' after "began" to remind the reader when the high-frequency measurements began.

line 185 - is that a reduction in emissions of CF4 per tonne of aluminium produced, or an actual reduction in emissions?

line 187 - The increase for CF4 between 1979 and 2019 is nearly 3 times, if you subtract off the natural background first. I believe that is the increase to compare to, but this should be made clearer in the text. It is not clear to me whether the authors took the increase for CF4 as close to 3 times, or less than 2 times (i.e. without subtracting the natural background) but "substantially larger" and "major sources of C2F6 not linked to the aluminium industry" sound big, suggesting the second option.

Figs 2-4 consider showing the global emissions for each PFC.

Line 211 - could spell out "sensitivity" of what to what, i.e. sensitivity of the measurement network to emissions from southern France and eastern Germany...

line 220 "no significant trend in NW Europe" - be more specific, is that no significant trend in NW European emissions?

line 225 - it looks to me like better agreement after 2016, rather than 2015

line 231 - "In the early and later parts of the record, this source is evident in our spatial maps" - the CF4 spatial map for 2005 in Fig 6 doesn't show any emissions around Dublin (2019 does) - should we expect to see emissions near Dublin on this map?

Line 335 - missing full stop

Fig 2 caption, last 2 lines - add "(dashed lines)" as follows: for each semi-hemisphere

[Figure]

(dashed lines) and ...
* * *

---

## Referee Comment (RC2) · Anonymous Referee #2 · 4 Dec 2020

Review for:

Title: "Global trends and European emissions of tetrafluoromethane (CF4), hexafluoroethane (C2F6) and octafluoropropane (C3F8)" Authors: Say et al.

Say et al. used the atmospheric measurements from AGAGE to infer global and European emissions of three PFCs (CF4, C2F6, C3F8). They further compare their regional top-down estimates and UNFCCC reports for emissions of these gases to assess the accuracy of national GHG reporting. The authors used well-established methods to conduct their analysis. The discussion is mostly well written. I would only suggest some clarification in their description of methods and some minor modifications.

[Figure]

Line 120: "the annual emissions growth rate would be zero plus or minus 20% of the maximum emissions...". What do you mean by "growth rates +- emissions"?

Lines 155 – 165: Could you please justify why it is reasonable to assume model uncertainty being "a combination of the prior baseline uncertainty and the magnitude of the median pollution event at the measurement location per year" and the correlation of the off-diagnoal elements in the model-data mismatch matrix is 12 hours?

Lines 215 – 216: the authors find no statistical significance in the emission trend of CF4 given the uncertainty of the estimated emissions. I am not sure I am convinced for this augment. Although the uncertainty seems relatively large in their derived annual emissions. They do show a clear reduction of emissions just visually. I would recommend the authors to calculate the trend and its statistical uncertainty. One way to do that could be to first generate hundreds of annual emissions sets by randomly sampling the errors of the annual emission estimates. Then you can conduct linear regression for those emission sets. It will yield a range of slopes / trends, which can be used for deriving the uncertainty of the slope.

Lines 225 – 230: the authors only mentioned Ireland's CF4 emissions have been increasing in recent years. How about Germany and France? Their posterior emissions also show indications of increasing emissions in recent years.

Line 258. Similar to what I suggested above, calculate the uncertainty of the emission trend over NW Europe, then assess if there is a statistically significant trend. Also, why the uncertainty in 2008 is much larger than other years? Is it because less observations in this year? If that is the case, maybe considering removing this year when constructing the trend and uncertainty.

Line 287: "C3F8 is the only PFC for which northwest Europe's. emissions increased relative to the global total." This statement seems contradicting with the previous sentence "northwest European emissions of C3F8 285 exhibited no statistically significant trend over the measurements period".

[Figure]

Lines 344 – 345: It is an interesting idea to suggest using C3F8 to be a transport tracer to evaluate transport errors. It would only work well if we know its emission magnitudes and distribution accurately. You mentioned about this in Section 3.2.4, but omitted it in this conclusion sentence.

---

## Author Comment (AC1) · 18 Dec 2020

We thank the two anonymous reviewers for their detailed and insightful feedback on our draft manuscript. Please find below a point-by-point response to the reviewers' comments and, where appropriate, a list of changes made.

Review 1

Reviewer: Line 13-14 "Unlike CF4 (and to a lesser extent C2F6), we observed no clear minimum associated with the 2008 financial crisis for C3F8." This sentence could be expressed more clearly to state what was observed for each of the 3 PFCs. It is

currently written in a bit of a backwards way.

Author response: We agree with the reviewer that the original sentence was poorly structured. Upon reflection, we do not feel that the signal associated with the 2008 financial crisis is a key finding of the study, not least because it has been presented and discussed previously. We therefore remove L13-14 from the abstract. References to the financial crash within the main text are retained.

Reviewer: Fig 2-4 I assume the symbols are monthly averages of baseline measurements, should "baseline" be mentioned in the figure captions?

Author response: The reviewer's assumption is correct. The captions for Fig. 2-4 now read: 'Monthly averaged baseline observations are shown as data points with $1\sigma$ error bars.'

Reviewer: line 182 - add "in 2002' after "began" to remind the reader when the high-frequency measurements began.

Author response: The reviewer proposes a useful addition. High-frequency measurements began in 2003. Hence, L183 now reads: 'At 14.1 Gg yr-1, emissions in 2018 are the largest observed since high-frequency measurements began in 2003.'

Reviewer: line 185 - is that a reduction in emissions of CF4 per tonne of aluminium produced, or an actual reduction in emissions?

Author response: The text refers to a decrease in CF4 emissions per ton of aluminium produced. This is clarified in the updated manuscript – L185 now reads: 'Despite this, given global primary aluminium production has increased $\sim$5-fold over the last 40 years (IAI, 2020), our estimates highlight the success of efficiency improvements and abatement technology in reducing emissions of CF4 per ton of aluminium produced.'

Reviewer: line 187 - The increase for CF4 between 1979 and 2019 is nearly 3 times, if you subtract off the natural background first. I believe that is the increase to compare to, but this should be made clearer in the text. It is not clear to me whether the authors took

the increase for CF4 as close to 3 times, or less than 2 times (i.e. without subtracting the natural background) but "substantially larger" and "major sources of C2F6 not linked to the aluminium industry" sound big, suggesting the second option.

Author response: We did not consider the natural background in the original manuscript, which on reflection was an oversight. As the reviewer correctly states, once this is considered, the increase in CF4 background is approximately 3-fold. We add this to the manuscript – L176 now reads: 'The global mean increased from 52.1 ppt in 1979 to 85.5 in 2019, representing an approximate 3-fold increase following subtraction of the natural background.' While there is still a clear discrepancy between gases – 3-fold for CF4 and 4-fold for C2F6, we remove 'substantially' from L189, reflecting that the difference is not as large as previously stated.

Reviewer: Figs 2-4 consider showing the global emissions for each PFC.

Author response: Estimated global emissions are shown in Table 2. We tested the inclusion of global emissions in Fig. 2-4, however found the resulting plots to be rather cluttered or too large. Therefore, we continue to present global emissions as a table only.

Reviewer: Line 211 - could spell out "sensitivity" of what to what, i.e. sensitivity of the measurement network to emissions from southern France and eastern Germany...

Author response: We define sensitivity on L210-211: '...based on the sensitivity of the measurements to emissions from these countries (see Fig. 1).' We have therefore chosen not to redefine sensitivity on L213, as to do so would risk sounding repetitive.

Reviewer: line 220 "no significant trend in NW Europe" - be more specific, is that no significant trend in NW European emissions?

Author response: We clarify our original statement by adding 'emissions' to L223-224, so that is now reads: 'Despite no significant trend in northwest Europe's emissions, global emissions increased considerably over the same period...'

[Figure]

Reviewer: line 225 - it looks to me like better agreement after 2016, rather than 2015

Author response: This comment refers to Fig. 5, which shows a comparison of our top-down CF4 estimates for NW Europe vs. UNFCCC data. The plot shows that, starting in 2015, the InTEM uncertainty bounds overlap the mean UNFCCC estimate, and hence we believe that our original statement was correct. However, to avoid any confusion, we have modified this statement so that L227-228 now reads: 'Our estimates are typically larger than the inventory, but some agreement is observed from 2016 onwards.'

Reviewer: line 231 - "In the early and later parts of the record, this source is evident in our spatial maps" - the CF4 spatial map for 2005 in Fig 6 doesn't show any emissions around Dublin (2019 does) - should we expect to see emissions near Dublin on this map?

Author response: Fig. 6 does show a very faint source centered on Dublin, however on reflection we feel that it is unlikely to be spotted and is therefore a potential source of confusion for the reader. To avoid this, we no longer refer to the spatial map for 2005. L234-235 now reads: 'From 2017 onwards, this source is evident in our spatial maps. It is therefore likely that emissions from this site are the main driver of the recent observed increase in Ireland's CF4 emissions.'

Reviewer: Line 335 - missing full stop

Author response: We add the missing full stop to the updated manuscript.

Reviewer: Fig 2 caption, last 2 lines - add "(dashed lines)" as follows: for each semi-hemisphere (dashed lines) and ...

Author response: We add '(dashed lines)' to the captions of Fig 2-4.

Review 2

Reviewer: Line 120: "the annual emissions growth rate would be zero plus or minus 20% of the maximum emissions. . .". What do you mean by "growth rates +- emis-

sions"?

Author response: We believe our original description to be accurate. In the current context, 'growth rate' refers to the change in emissions from year to year and is not linked to the growth rate discussed in section 3.1, which refers to growth in the atmospheric mole fraction (note that the text says 'emissions growth rate'). In short, growth in emissions between any two years is constrained to be zero plus/minus 20% of the maximum emissions (in any year) reported to the EDGAR database.

Reviewer: Lines 155 – 165: Could you please justify why it is reasonable to assume model un- certainty being "a combination of the prior baseline uncertainty and the magnitude of the median pollution event at the measurement location per year" and the correlation of the off-diagonal elements in the model-data mismatch matrix is 12 hours?

Author response: Defining model uncertainty is extremely difficult and there is no absolute way of doing this. The removal of selected observations when the model atmosphere is estimated to be stable is the first stage in attempting to ensure the model is performing acceptably for the remaining periods. The next stage is to provide a mole fraction uncertainty for each measurement. The model timing of any pollution event can be off by several hours, and the model boundary layer height (and thus atmospheric mixing) at any time can be poorly estimated so the uncertainty for any particular event has to reflect this. Taking the median of the magnitude of the pollution events over a year is an attempt, albeit arbitrary, to help reflect the potential size of the uncertainty. Another key component is using the variability of the observations over the 12-hr window centered on the particular 4-hr InTEM time-window. The fitting of the baseline can also lead to over or under-representing the pollution events hence the quality of the baseline fit at each time is important to reflect in the overall uncertainty. The list of references given in the text (particularly Manning et al., 2011) describe the estimation of model uncertainty. An e-folding time correlation in the uncertainty covariance implies that model-measurement uncertainties should be considered correlated (i.e.

not independent) over some timescale (12 hours in this case). The sensitivity of the results to this arbitrary value is low for reasonable changes. Previous tests with 8- and 16-hour windows show that there is little discernible impact on the annually estimated national emissions. We add this information to the manuscript – L160-164 now read: 'Off-diagonal elements of the model-observation uncertainty matrix were calculated by assuming a temporal correlation timescale of 12 hours. The sensitivity of the results to this arbitrary value is low for reasonable changes - tests with 8- and 16-hour windows showed that there was little discernible impact on the annually estimated national emissions.'

Reviewer: Lines 215 – 216: the authors find no statistical significance in the emission trend of CF4 given the uncertainty of the estimated emissions. I am not sure I am convinced for this augment. Although the uncertainty seems relatively large in their derived annual emissions. They do show a clear reduction of emissions just visually. I would recommend the authors to calculate the trend and its statistical uncertainty. One way to do that could be to first generate hundreds of annual emissions sets by randomly sampling the errors of the annual emission estimates. Then you can conduct linear regression for those emission sets. It will yield a range of slopes / trends, which can be used for deriving the uncertainty of the slope.

Author response: Thank you for this suggestion. We have estimated the linear emissions trend and its uncertainty using an MCMC framework, under no prior assumption about the trend (nor intercept). For NWEU emissions of CF4, the 95% uncertainty interval includes both positive and negative trend values (-10.9 (-26.7 – 6.34) Mg yr-1), on which we base this statement that there is no significant trend. We have updated the text to reflect that this significance is based on the 95% uncertainty in the trend. L217-219 now read: 'Owing to the considerable uncertainties, we find no statistically significant trend in emissions from the region over the measurement period (based on the 95% uncertainty in the trend).'

Reviewer: Lines 225 – 230: the authors only mentioned Ireland's CF4 emissions have

been in- creasing in recent years. How about Germany and France? Their posterior emissions also show indications of increasing emissions in recent years.

Author response: See previous response for full details. We used a simple MCMC framework to estimate the uncertainty in the trend for each country/region. While there does appear to be a small uptick in emissions from Germany and France from 2017 onwards, the uncertainty in the trend for these years included both positive and negative values (based on the 95% uncertainty), suggesting that there is no statistical significance.

Reviewer: Line 258. Similar to what I suggested above, calculate the uncertainty of the emission trend over NW Europe, then assess if there is a statistically significant trend. Also, why the uncertainty in 2008 is much larger than other years? Is it because less observations in this year? If that is the case, maybe considering removing this year when constructing the trend and uncertainty.

Author response: The larger uncertainty is indeed due to fewer observations in 2008. Measurements from the Jungfraujoch station began in April 2008. Prior to this, esti-mates are based solely on observations from Mace Head, resulting in larger uncertain-ties for countries furthest from this site (e.g., Germany). Using our MCMC framework (full details above), we estimate the NWEU trend to be -1.54 (-5.57 – 2.82) Mg yr-1. As the trend uncertainty includes both positive and negative values, we can state that there is no statistical significance. We add this to L259-61, so that they now read: 'Like CF4, the uncertainty of our estimates for Germany and France, particularly in early years, is large and overall, there is no statistically significant trend in emissions from northwest Europe over the measurement period (based on the 95% uncertainty in the trend).'

Reviewer: Line 287: "C3F8 is the only PFC for which northwest Europe's. emissions increased relative to the global total." This statement seems contradicting with the pre-vious sentence "northwest European emissions of C3F8 285 exhibited no statistically

significant trend over the measurements period".

Author response: While our top-down results suggest that northwest European C3F8 emissions have not increased/decreased over the measurement period, global emissions have decreased (0.69 Gg in 2008, 0.55 Gg in 2018). Therefore, as a percentage of the global total, northwest Europe's emissions have increased. However, we agree that the sentence requires some clarification to avoid any confusion for the reader. We update L289-290 so that they now reads: 'Of the gases studied, C3F8 is the only PFC for which northwest Europe contributed a greater fraction of the global total in 2018 than it did in 2008.'

Lines 344 – 345: It is an interesting idea to suggest using C3F8 to be a transport tracer to evaluate transport errors. It would only work well if we know its emission magnitudes and distribution accurately. You mentioned about this in Section 3.2.4, but omitted it in this conclusion sentence.

Author response: Further work is required to fully evaluate the possibility of using C3F8 as a tracer and as the reviewer asserts, it will depend on our ability to fully understand both the distribution and magnitude of emissions. However, our top-down analysis does indicate that European C3F8 emissions are dominated by a single source. Likewise, the forward model suggests that the majority of observed events can be explained by a source in North West England. The conclusion does state that high frequency data would be required for the tracer experiment to be a success. Nevertheless, we clarify this by modifying L346-348, which now reads: 'We explored the potential of using this facility in a tracer release experiment and showed that, if accurate high frequency emissions data were made available, this site could be used to provide useful information related to transport model performance.'
* * *